# Regulation of the Cancer Stem Phenotype by Long Non-Coding RNAs

**DOI:** 10.3390/cells11152352

**Published:** 2022-07-30

**Authors:** Jose Adan Gutierrez-Cruz, Vilma Maldonado, Jorge Melendez-Zajgla

**Affiliations:** 1Functional Genomics Laboratory, Instituto Nacional de Medicina Genomica, Mexico City 04710, Mexico; joseagc992@gmail.com; 2Epigenetics Laboratory, Instituto Nacional de Medicina Genomica, Mexico City 04710, Mexico; vmaldonado@inmegen.gob.mx

**Keywords:** stem phenotype, long non-coding RNA, microenvironment, stem cancer cells

## Abstract

Cancer stem cells are a cell population within malignant tumors that are characterized by the ability to self-renew, the presence of specific molecules that define their identity, the ability to form malignant tumors in vivo, resistance to drugs, and the ability to invade and migrate to other regions of the body. These characteristics are regulated by various molecules, such as lncRNAs, which are transcripts that generally do not code for proteins but regulate multiple biological processes through various mechanisms of action. LncRNAs, such as HOTAIR, H19, LncTCF7, LUCAT1, MALAT1, LINC00511, and FMR1-AS1, have been described as key regulators of stemness in cancer, allowing cancer cells to acquire this phenotype. It has been proposed that cancer stem cells are clinically responsible for the high recurrence rates after treatment and the high frequency of metastasis in malignant tumors, so understanding the mechanisms that regulate the stem phenotype could have an impact on the improvement of cancer treatments.

## 1. Introduction

Cancer is one of the leading causes of death worldwide. According to the World Health Organization, in 2020, there were an estimated 18.1 million new cases and 9.9 million deaths worldwide. The most frequently diagnosed malignant tumors are breast, lung, colorectal, prostate, and stomach cancers, while the leading causes of death are lung, breast, colorectal, liver, and stomach cancers [1]. The main causes of death from these diseases are recurrence after treatment [2] and the development of metastases [3], which are caused by the appearance of aggressive cell populations such as cancer stem cells [4,5]. These cells are a subpopulation of malignant tumor cells that possess the ability to self-renew [6], differentiate into non-stem cancer cells that are part of the heterogeneity of the malignant tumor [7], and evade many of the drugs directed against proliferating cancer cells [8]. Stem cells or tumor-initiating cells (TICs) express receptors, transcription factors, and microRNAs that define their identity [9,10] and enhance their ability to invade and migrate to other regions of the body to form new malignant tumors [11]. Among the molecules involved in the regulation of stemness or stem phenotype are long non-coding RNAS (lncRNAs), which are transcripts larger than 200 nucleotides that, for the most part, do not have an open reading frame [12] but regulate multiple biological processes, such as cell proliferation [13], cell death [14], differentiation [15], and cell homeostasis [16]. These effects are mediated by several mechanisms, including acting as scaffolds for proteins complexes, decoys for RNAs, sponges for microRNAs, and guides for transcription factors. In this review, we describe the relationship between lncRNAs and the acquisition of the stem phenotype by cancer cells, as well as the biological implications of this relationship.

## 2. Cancer Stem Cells

Malignant tumors are composed of heterogeneous cell populations [17,18] with different genetic [19], epigenetic [20], and phenotypic [21] alterations. One of the cell populations that has attracted the attention of scientists in recent years is cancer stem cells. These cells are a cell population within malignant tumors that possess several characteristics (Figure 1) that, at the clinical level, are responsible for recurrence after surgery or treatment with antineoplastic drugs and radiotherapy [22,23], in addition to the high rate of metastasis generated in many cancers [24], which are the main causes of death in most malignant tumors [3]. Cancer stem cells (CSC) represent an interesting target for therapy, as they are responsible for these clinical features, and their elimination could result in increased survival rates and even eradication of the disease. Since their discovery in the 1990s by John Dick’s work group in acute myeloid leukemia [4], there have been numerous studies in which populations of these cancer cells have been reported. To date, cancer stem cells have been identified in breast cancer [5,25,26], colorectal cancer [27,28,29,30,31], stomach cancer [32], lung cancer [33], and prostate cancer [34], among other malignant tumors.

Cancer stem cells can originate from normal tissue stem or progenitor cells that have accumulated genetic and epigenetic alterations, causing their transformation to cancer cells [35]; however, it has also been proposed that cancer cells can acquire a stem phenotype in response to epigenetic modifications and stimuli generated by the tumor microenvironment [36].

### Characteristics of the Stem Phenotype

The stem phenotype is defined by a series of characteristics that cancer cells acquire during the development of the malignant tumor [37]. These characteristics are similar to those possessed by normal stem cells that are found at the base of every tissue in our body [38]; hence, one of the proposals to explain the origin of cancer stem cells is that of the conversion of a normal stem cell that has acquired genetic and epigenetic alterations that lead to its conversion to a cancer stem cell [39]. The main characteristics that define the cancer stem phenotype are described below:Self-renewal. This is the ability of cancer stem cells to divide asymmetrically [40], causing one of the daughter cells resulting from cell division to maintain its stem characteristics [41] and the other resulting daughter cell to begin the process of a restricted differentiation program without the stem cell capabilities [42]. Cancer stem cells maintain their population at a constant level through this feature, allowing the malignant tumor to reform when, for example, non-stem cancer cells are pharmacologically eliminated [43].Tumorigenic ability. Cancer cells with a stem phenotype possess the ability to initiate and maintain a malignant tumor when transplanted into immunodeficient mice [4,5]. A small number of them is sufficient to recapitulate the malignant tumor from which they originate when transferred in this type of mouse [44,45], in contrast to non-stem cancer cells, which present a reduced tumor-forming ability [46].Drug resistance. Cancer stem cells are able to evade many conventional chemotherapeutic drugs [47,48]. These cells can remain quiescent for long periods of time, so drugs directed toward proliferating cells do not affect them, as they are not dividing [49]. Another mechanism involved in their drug resistance is the expression of ATP-dependent membrane transporters of the ABC family [50] or isoforms of aldehyde dehydrogenase enzymes, which allow the expulsion of drugs more efficiently than in cancer cells that do not express these proteins [51].Marker expression: The expression of specific molecules is one of the features that has allowed the identification, isolation and enrichment of cancer stem cells. The markers range from surface proteins, such as CD44 [32], CD133 [33], CD24 [5], EpCAM (*epithelial cell adhesion molecule*) [52], etc., to molecules with enzymatic activity, such as isoforms 1A1 and 1A3 of the enzyme aldehyde dehydrogenase (ALDH) [25,26]. In general, cancer stem cells usually express some of the transcription factors associated with pluripotency, such as SOX2, NANOG and OCT4) [53,54]. Table 1 lists the main markers that define the stem phenotype in the most common solid tumors.

**Table 1 cells-11-02352-t001:** Major cancer stem cell markers in the most common malignant tumors.

Markers	Tumor	References
CD44^+^ CD24^−^/low, CD44^+^ CD24^−^, CD44^+^ CD24^−^ CD24 EpCAM^+^, ALDH enzyme activity (isoforms 1A1, 1A3)	Breast cancer	[5,25,26]
CD133, CD166, CD44, LGR5, ALDH1A1 enzyme activity	Colorectal cancer	[27,28,29,30,31,32,33]
CD44^+^	Stomach cancer	[32]
CD133^+^	Lung cancer	[33]
CD133^+^, CD44^+^	Prostate cancer	[33,34]
CD44^+^ CD24^+^ EpCAM^+^, CD133^+^ CD44^+^, ALDH enzyme activity.	Pancreatic cancer	[52,55,56]
CD133^+^	Brain cancer	[57]

Metastasis. It has been proposed that cancer stem cells are able to migrate to other regions of the body and form new malignant tumors (metastasis) by activating the EMT (epithelial-mesenchymal transition) program that allows the conversion of an epithelial cell to a mesenchymal cell [11,58]. To perform this process, cancer stem cells present activated signaling cascades that regulate transcription factors such as SNAIL (*snail family transcriptional repressor 1*), SLUG (*snail family transcriptional repressor 2*), ZEB1 (*zinc finger E-Box binding homeobox 1*), and ZEB2 (*zinc finger E-Box binding homeobox 2*) [59,60,61,62].

Cancer cells possess highly dynamic phenotypic plasticity. This plasticity allows these cells to acquire stem characteristics in response to microenvironmental stimuli [63] such as hypoxic conditions [64], absence of nutrients (metabolic reprogramming) [65], cytokines secreted by the cancer cells themselves or cells of the immune system [66], and selection pressures such as the use of drugs [67]. The acquisition of the stem phenotype makes these cells difficult to eliminate, as it confers capabilities that favor their survival and thus progression to more aggressive stages of cancer in patients [68]. At the molecular level, the stem phenotype is regulated by multiple molecules that interact with each other to maintain this state in cancer cells. Among the signaling systems associated with the maintenance of stemness in cancer are the WNT/β-catenin [69], Notch [70], Sonic Hedgehog [71], and Hippo YAP/TAZ [72] signaling pathways, in addition to the activity of transcripts such as miRNAs [73] and lncRNAs [74].

## 3. LncRNAs Regulate Gene Expression

Long non-coding RNAs (lncRNAs) are transcripts longer than 200 nucleotides that generally do not code for functional proteins [75]. Most of these transcripts are synthesized by RNA polymerase II [76] and, like many of the RNAs described to date, have a long adenine sequence at the 3′ end and a cap at the 5′ end (7 methyl guanosine) [77]. lncRNAs regulate multiple biological processes, such as proliferation [78], differentiation [79], cell death, etc. [80]. To date, more than 28,000 lncRNAs have been described in humans (according to the FANTOM5 project); however, we only know the function of some of them [81] (Figure 2).

LncRNAs can be transcribed from different regions of the genome and, according to their location in the genome, they are classified into sense lncRNAs, when transcribed from the sense strand of a coding gene (overlapping with exons or introns) [82]; antisense lncRNAs, located on the antisense strand of a coding gene [83]; intronic lncRNAs, produced from intronic regions of coding genes [84]; bidirectional lncRNAs, formed on the opposite strand and direction of a coding gene [85]; enhancer lncRNAs, which are derived from genome sequences that act as enhancers [86]; and intergenic lncRNAs, located in regions located between coding genes [87]. LncRNAs regulate multiple processes for cell homeostasis, so their mechanisms for carrying out this work are also varied. These RNAs can be grouped according to their molecular mechanism of action, into the categories listed below:Scaffolds: lncRNAs can form secondary structures that allow them to bind to proteins to form complexes that carry out particular functions in the cell [88]. An example of a transcript with this mechanism of action is lncRNA-NEAT1 (*nuclear paraspeckle assembly transcript 1*), which interacts with the protein subunit EZH2 (*enhancer of zeste 2 polycomb repressive complex 2 subunit*) of the chromatin remodeling complex PCR2 (*polycomb repressive complex 2*), which functions as a transcriptional repressor of different genes [89] (Figure 2).Sponges: lncRNAs with this mechanism of action present target miRNAs sequences, so they can prevent these miRNAs from carrying out their function by directly interacting with them. The miRNAs are complementary to the lncRNA sequence [90]. Among the transcripts with this mechanism of action is lncRNA-SNHG7 (*small nucleolar RNA hostgene 7*), which is found abundantly in the cell and hijacks miR-216b, blocking its activity by competing as a binding site against its target messenger RNA (mRNA) [91] (Figure 2).Guides: lncRNAs can interact with chromatin remodeling proteins and transcription factors to target specific DNA sequences where gene activation or silencing is required [92]. One of the best studied lncRNAs with this function is HOTAIR (*HOX transcript antisense RNA*), which mediates the silencing of tumor suppressor genes such as P21 in cancer. In this case, HOTAIR recruits the chromatin remodeling complex PRC2, through its interaction with the EZH2 subunit, to the promoter region of P21, establishing marks that allow the formation of heterochromatin and, therefore, the transcriptional repression of the tumor suppressor gene [93] (Figure 2).Signals: Acting as markers in different regions of the genome to signal chromatin remodeling complexes whether transcriptional activation or silencing of genes is required [92]. One of the lncRNAs reported to have this function is XIST (*X inactive specific transcript*). When this transcript is expressed, it functions as a mark indicating which X chromosome should be silenced, in females. XIST recruits chromatin remodelers, such as PRC2, to the X chromosome that will be repressed to generate heterochromatin [94] (Figure 2).Decoys: Preventing proteins or protein complexes such as the transcription initiator complex from carrying out their function, thus impacting gene expression. LncRNAs with decoy activity bind to proteins and DNA, preventing other molecules from interacting with them. In this way, the hijacked protein or DNA sequence cannot interact with its target. The lncRNA TERRA (*telomeric-repeat-containing RNA*) is an example of a transcript with this mechanism of action. This lncRNA binds to the active site of the enzyme telomerase, blocking its activity and preventing it from carrying out its function of protecting the telomeric regions of chromosomes [95] (Figure 2).

## 4. The Role of LncRNAs in the Regulation of the Stem Phenotype in Malignant Tumors

Alteration of lncRNAs expression participates in the development of diseases such as cancer [96]. Their oncogenic or tumor suppressor activity has been associated with all the characteristics or hallmarks that define this group of diseases [97,98]. The stem phenotype is no exception since, in recent years, several research groups around the world have described the role that lncRNAs play in the acquisition and maintenance of this phenotype. Most of the research has been performed in a limited number of tumor types, including breast and gastrointestinal tumors. The most relevant examples of lncRNAs related to stemness in different malignancies are described below.

### 4.1. Breast Cancer

Globally, breast cancer is the malignant tumor with the highest incidence and the second highest mortality in women [1]. These tumors are highly heterogeneous, and different cell populations that contribute to their development and maintenance have been described [99]. Among the cell populations described in these malignant tumors are cancer stem cells, which were first described in 2003 by Al-Hajj et al. [5]. Their group demonstrated that cancer cells with the CD44^+^ CD24^−/low^ phenotype had a greater tumorigenic capacity when transplanted into immunodeficient mice (NOD/SCID), compared to those cells that did not have this phenotype. Over time, other markers associated with stemness in these malignancies have been described, such as the expression and activity of the A1A1 and A1A3 isoforms of the ALDH enzyme [25,26]. Among the molecules involved in the regulation of the stem phenotype in breast cancer are lncRNAs (Figure 3). In 2021, Ten et al. reported that Lnc408 was able to regulate the stem phenotype in breast cancer through modulation of the β-catenin protein, which is part of the WNT signaling pathway responsible for regulating stemness in different malignancies [100,101]. This research group demonstrated that, when Lnc408 is expressed, it recruits the transcription factor SP3 to the promoter region of the CBY-1 (*protein chibby homolog 1*) protein to prevent its expression. CBY-1 acts by inhibiting β-catenin, by interacting directly with it and promoting its degradation. When CBY-1 is absent, β-catenin is not degraded and can perform its activity, activating the WNT signaling pathway. The activation of this signaling pathway promotes the expression of the stem transcription factors SOX2 and NANOG, as well as CD44, which is one of the reported stem markers in breast cancer. This allows cancer cells to acquire a greater ability to form malignant tumors when transplanted into immunodeficient mice. HOTAIR is an additional lncRNA with a similar mechanism of action to Lnc408 [102]. This transcript regulates the expression of miR-304a in cancer stem cells of both MCF-7 and MDA-MB-231 cell lines. HOTAIR increases the expression of the transcription factor SOX2 by epigenetically silencing the expression of this miRNA, which normally negatively regulates the mRNA of this transcription factor. HOTAIR expression in these cell models allows cancer stem cells to self-renew and thus maintain constant populations throughout cell divisions [102]. Microenvironmental conditions play a relevant role in the acquisition of the stem phenotype in cancer, as it has been reported that conditions such as hypoxia allow cancer cells to acquire stemness characteristics [103]. Interestingly, lncRNAs are also involved in this process. In 2021, Zhu et al. reported that hypoxia induces the expression of lncRNA-KB-1980E6.3, which interacts with IGF2BP1 *(**insulin-like growth factor 2 mRNA-binding protein 1*) protein to form an RNA–protein complex with the C-MYC oncogene mRNA and stabilize it [104]. The formation of this complex promotes the translation of the oncogene mRNA, causing cancer cells to acquire increased malignant tumor-forming capacity in vivo and increased protein levels of pluripotency factors OCT4, SOX2, NANOG and KLF4 (*Kruppel-like factor 4*). LncRNAs also interact with miRNAs by directly binding to their nucleotide sequence, sequestering them, and preventing them from carrying out translation inhibition of their target mRNA (Figure 2). LUCAT1 is a type of lncRNA that possesses this mechanism of action. When the expression of this transcript increases, it acts as a sponge for miR-5582-3p, which is a negative regulator of the mRNA transcription factor TCF7L2. The presence of LUCAT1 allows the transcription factor mRNA to be translated into protein which activates the WNT signaling pathway, resulting in cancer cells with stem characteristics [105]. Even though the molecular mechanisms by which lncRNAs regulate stemness have been explored for just a few of them, a relationship between lncRNA expression and the stem phenotype has been clearly established by loss- or gain-of-function assays for a larger number of lncRNAs. For example, this is the case with the lncROPM lncRNA, which is expressed in the CD44^+^ CD24^−^/low breast cancer stem cell population. When this lncRNA is inhibited, it decreases the drug resistance, self-renewal, invasiveness, and metastatic capacity of these cells [106]. The list of lncRNAs associated with the regulation of the stemness phenotype in breast cancer continues to grow and the lncRNAs described as regulators of stemness in these malignancies include LINC00617 [107], LncRNA-AGAP2-AS1 [108], lncRNA-HAL [109], LINC00511 [110], LncRNA-H19 [111], LncRNA-HOTTIP (*HOXA*
*distal transcript antisense RNA*) [112], LncRNA-FEZF1-AS1 [113], LINC00261 [114], LncRNA-Hh [115], LncRNA-SOX21-AS1 [116], LncRNA-NEAT1 [117], LincRNA-ROR (*regulator of reprogramming*) [118], and MALAT-1 (*metastasis associated lung adenocarcinoma transcript* 1) [119], among others. Table 2 describes the role of these lncRNAs in the regulation of stemness in breast cancer. Clearly, more research is needed to explore the molecular mechanisms used by lncRNAs to accomplish their regulatory role. The use of new high-throughput techniques such as CRISPR-Cas high-content screening should help to achieve this [120].

### 4.2. Colorectal Cancer

Colorectal cancer has the second highest incidence and third highest mortality worldwide [1]. These malignant tumors originate in the crypts of the colon, and cancer cell populations with a stem phenotype have also been reported in these malignancies. In 2007, O’Brien et al. reported the presence of CD133^+^ cells with a high tumor-forming capacity when transplanted into immunodeficient (NOD/SCID) mice [27]. This was the first time that CSCs were detected in colorectal cancer. In recent years, different cancer stem cell populations have been reported in colon cancer, as defined by the markers LGR5^+^, CD44^+^, and ALDH positive activity, among others [28,30,31]. LncRNAs have been also described as key regulators of stemness in these malignancies (Figure 4). One of the best described transcripts in this context is H19. This is a lncRNA of approximately 2300 bases located on chromosome 19 that has been associated with different oncogenic processes, such as uncontrolled cell proliferation and metastasis, in many malignant tumors. In colorectal cancer, H19 regulates the stemness of cancer cells with ALDH enzyme activity. When this lncRNA is inhibited, the capacity of tumor formation in immunodeficient mice (NOD/SCID) decreases, as does the resistance to oxaliplatin (a drug frequently used as chemotherapy in this type of cancer) and the proportion of ALDH cells^+^ [121]. H19 regulates these processes by acting as a sponge, sequestering miR-141, which normally negatively regulates β-catenin mRNA. Upon overexpression of H19, miR-141 activity is prevented, allowing translation of β-catenin mRNA, and activating the WNT signaling pathway [121]. LncRNAs are able to modulate key pluripotent signaling systems that control stemness in cancer, such as WNT and secondary or restricted stem-associated signaling pathways, such as Sonic Hedgehog. In 2019, Zhou et al. showed that inhibition of lncRNA-cCSC1 decreased the levels of SMO and Gli1 proteins, intermediates of the Hedgehog system [122]. This negative modulation of the signaling pathway decreased in vitro invasion and migration, the protein levels of CD44 and CD133 markers, and the tumor-forming ability of cancer cells in immunodeficient mice (BALB/c), although the exact molecular mechanism by which lncRNAs regulate the Hedgehog pathway remains unknown [122]. Among the processes involved in the maintenance of cancer stem cell populations in malignant tumors is the ability of these cells to divide asymmetrically. After a cancer stem cell divides, one of the resulting cells begins the process of partial differentiation, while the other resulting cell maintains its stem characteristics. The distribution of molecules after division is different in each of these cells. An example lncRNA regulating this process is Lnc34a, which is expressed in cancer cells with a stem phenotype but not in cancer cells that have begun the differentiation process [123]. Lnc34a allows cancer stem cells to self-renew by recruiting chromatin remodeling proteins, such as HDAC1 and DNMT3a (via PHB2), to the promoter region of miR-34a to prevent its expression. Inhibition of this miRNA by Lnc34a allows cancer stem cells to maintain their identity through cell divisions. Additional lncRNAs that have been described as regulators of stemness in colorectal cancer are lncRNA-TUG 1 [124], lncRNA-BCAR4 [125], lncRNA SLCO4A1-AS1 [126], and lncRNA KLK8 [127].

### 4.3. Gastric Cancer

The tumor microenvironment plays an important role in the acquisition of the stem phenotype in cancer [128]. Microenvironment cells (cancerous or otherwise) secrete molecules, such as cytokines, that modify the phenotype of cancer cells [129]. These molecules modulate signaling cascades that culminate in the activation of genes associated with stemness regulation, such as genes encoding for lncRNAs. In gastric cancer, this phenomenon has been widely reported. In 2019, He et al. reported that mesenchymal stem cells in the tumor microenvironment secrete TGF-β1 (*transforming growth factor beta 1*), which induces the activation of SMAD 2/3 proteins in cancer cells, which promote the expression of lncRNA-MACC1-AS1 [130]. This lncRNA acts as a sponge for miR-145-5p, which negatively regulates mRNAs associated with fatty acid metabolism in the cell. Sequestration of miR-145-5p by lncRNA-MACC1-AS1 induces the expression of stemness markers CD133, OCT4, and SOX2, and increases both oxaliplatin resistance and tumorigenic capacity of cancer cells [130]. Interestingly, lncRNAs can also be secreted by cells from the microenvironment through exosomes and act in as cell-autonomous factors by regulating the stemness phenotype. This is the case for LINC01559, which is secreted by mesenchymal stem cells via exosomes into the tumor microenvironment [131]. Exosomes containing LINC01559 are received by cancer cells, where the lncRNA allows these cells to acquire a greater tumorigenic capacity, the ability to migrate in vitro, and pharmacological resistance. This is achieved through the regulation of the PI3K/AKT signaling pathway. Molecularly, LINC05199 acts as a sponge, sequestering miR-1343-3p and promoting PGK1 (*phosphoglycerate kinase 1*) mRNA translation into its protein form. The kinase activates the PI3K/AKT signaling pathway, allowing cancer cells to acquire stem-like characteristics. On the other hand, LINC05199 can translocate to the cell nucleus and recruit chromatin remodelers (EZH2) to the promoter region of the phosphatase PTEN (*phosphatase and tensin homolog*), blocking its expression. This phosphatase is a negative regulator of the PI3K/AKT pathway, so its inhibition leads to the activation of this signaling pathway (Figure 5). Some lncRNAs, such as lncRNA-FEZF1-AS1, regulate the stem phenotype in several tumor types. This is the case for gastric and breast cancer, where the altered expression of this lncRNA results in the regulation of the transcription factors SOX2, OCT4, and NANOG [113,132]. The number of lncRNAs involved in the regulation of cancer stemness points toward a complex signaling redundancy and supports a wider role of these RNAs in the stem phenotype of normal cells.

### 4.4. LncRNAs in the Regulation of the Stem Phenotype of Other Types of Cancer

LncRNAs are important modulators of the stem phenotype in the most common malignancies, including leukemias. CSC are key players in leukemia initiation, progression, and resistance to therapy [133] and lncRNAs also play important roles in maintaining the stem phenotype of these cells. For example, it has been shown that the lncRNA HOTTIP is not only overexpressed in acute myeloid leukemia (AML) patients but is key in maintaining stemness via regulation of the HOXA gene [134]. Additional hematopoietic stem factors are also regulated by lncRNAs in leukemias, such as TET2 and TET3 by MAGI2-As3, HOXA10-AS, and WT1 non-coding RNAs [135,136]. As expected by their targets, these lncRNAs are required for the stem phenotype maintenance, as loss-of-function decreases the number of leukemia stem cells in vivo. More importantly, a therapeutic role for stem-associated lncRNAs can be envisioned. DANCR is a lncRNA that is overexpressed in cytogenetically normal AML patients, with the highest stem subpopulation levels. In vivo experiments showed that DANCR knock-down resulted in decreased stem cell renewal, pointing toward a possible use of this gene for therapy [137].

There are also studies that report the regulation of this phenotype in less common or less investigated malignancies. In 2021, lncRNA-MEG3 was reported to act as a tumor suppressor in oral cancer [138]. Chen and coworkers showed that this lncRNA negatively modulates the WNT signaling pathway by sequestering miR-421 and preventing its activity. Interestingly, lncRNA-MEG3 has also been described as a tumor suppressor in stomach cancer, negatively regulating the JAK/STAT3 signaling pathway by sequestering a different miRNA, miR-708 [139]. In both cancers, the lncRNA reduces the tumor-forming capacity of cancer stem cells when transplanted into immunodeficient mice. Another transcript that regulates stemness in different tumors is LncTCF7. This is a transcript that, in liver cancer, works as a guide to recruit the SWI/WNF chromatin remodeling complex to the promoter region of the TCF7 transcription factor in order to promote its expression. This activates the WNT signaling pathway, allowing cancer stem cells to self-renew [140]. In brain cancer, LncTCF7 negatively regulates miR-200c by sequestering it and inhibiting its activity [141]. This miRNA negatively regulates the mRNA of the stemness marker EpCAM, so its inhibition by LncTCF7 promotes the translation of this mRNA, activating the transcription factors SOX2 and NANOG. At the phenotypic level, this allows cancer stem cells to self-renew and have a greater tumorigenic capacity than non-stem cancer cells. Additional lncRNAs, such as HOTAIR, can act by regulating the stem phenotype in three different malignancies (breast, prostate and pancreatic) through similar mechanisms [102,142,143] (Table 2).

LncRNAs have been shown to be extremely versatile molecules in terms of mechanisms of regulation of the stem phenotype, so the alteration of their activity and their effect on this phenotype are still being described. Additional lncRNAs associated with the stem phenotype in cancer include GAS5 (*growth arrest specific* 5) [144], LncRNA-RP11-567G11.1, LncRNA-UC.345, and LncRNA STXBP5-AS1 (*STXBP5 antisense RNA 1*) in pancreatic cancer; LncRNA HAND2-AS1 (*heart and neural crest derivatives expressed transcript 2 antisense RNA 1*) [145] in liver cancer; LncRNA-UCA1 (*urothelial carcinoma associated 1*) in cervical cancer; and FMR1-AS1 (*FMR1 antisense RNA 1*) [146] in esophageal cancer. Table 2 describes the relationship between these lncRNAs and the stem phenotype.

**Table 2 cells-11-02352-t002:** LncRNAs that regulate the stem phenotype in the most common cancers.

LncRNA	Genes/Signaling Pathway Involved	Tumor	References
LncRNA-AGAP2-AS1	SOX2, OCT4	Breast	[108]
Lnc408	CBY-1/WNT signaling pathway	Breast	[147]
LncRNA-HOTTIP	miR-148a-3p/Wnt pathway	Breast	[112]
LncRNA-ROPM	Not addressed	Breast	[106]
LINC00617	CD44^+^ CD24^−^	Breast	[107]
lncRNA-HAL	CD44^+^ CD24^−^ via NANOG and ALDH1A3.	Breast	[109]
LncRNA-KB-1980E6.3	IGF2BP1-c-Myc	Breast	[104]
HOTAIR	miR-34a-SOX2	Breast	[102]
	miRNA-590-sp-JAK/STAT3 signaling pathway	Prostate	[142]
	miR-34a-JAK2/STAT3	Pancreas	[143]
LncRNA-FEZF1-AS1	miR-30a-NANOG	Breast	[113]
	miR-363-3p-SOX2, NANOG and OCT4	Stomach	[132]
LINC00261	miRNA-550a-3p	Breast	[114]
LUCAT1	miR-5582-3p-TCF7L2/WNT signaling pathway	Breast	[105]
LncRNA-Hh	Not addressed	Breast	[115]
LncRNA-SOX21-AS1	Hippo YAP/TAZ signaling pathway	Breast	[116]
LncRNA-NEAT1	SOX2	Breast	[117]
LincRNA-ROR	EMT program (ZEB1, ZEB2, TWIST, SLUG and SNAIL)	Breast	[118]
LncRNA-H19	let-7	Breast	[111]
	miR-141-β-catenin	Colorectal	[121]
LINC00511	miR-185-3p-E2F	Breast	[110]
MALAT-1	SOX2	Breast	[119]
	Bmi1, β-catenin, C-MYC, NANOG, SOX2 and OCT4.	Pancreas	[148]
LNCRNA-TUG 1	Not addressed	Colorectal	[124]
LncRNA-BCAR4	miR-665	Colorectal	[125]
LncRNA SLCO4A1-AS1	miR-150-3p-SLCO4A1	Colorectal	[126]
LncRNA-cCSC1	SMO/GLI1 (Hedgehog signaling pathway)	Colorectal	[122]
LncRNA KLK8	SOX2, OCT4 and NANOG	Colorectal	[127]
Lnc34a	miR-34a-DNMT3a/HDAC1	Colorectal	[123]
LncRNA-RP11-567G11.1	JAGGED1, HES1 and HES5 (Notch signaling pathway)	Pancreas	[149]
GAS5	miR-221-SOCS3	Pancreas	[144]
LncRNA STXBP5-AS1	EZH2-ADGB	Pancreas	[150]
LncRNA-uc.345	SOX2, OCT4 and NANOG	Pancreas	[151]
LINC01559	miR-1343-3p-PTEN-PI3K/AKT signaling pathway	Stomach	[131]
LncRNA-MACC1-AS1	TGF-β signaling pathway-miR-145-5P	Stomach	[130]
LncRNA-MEG3	miR-708-SOCS3-JAK/STAT3 pathway	Stomach	[139]
	miR-421-WNT signaling pathway	Oral	[138]
FMR1-AS1	NF-κB/c-Myc signaling pathway	Esophagus	[146]
LncRNA HAND2-AS1	INO80 chromatin remodeling complex-BMPR1A-BMP signaling pathway	Liver	[145]
LncTCF7	SWI/SNF chromatin remodeling complex-TCF7-WNT signaling pathway	Liver	[140]
	miR-200c-EpCAM	Brain (Glioma)	[141]
LncRNA-UCA1	miRNA-122-5p-SOX2	Cervical	[152]

## 5. Conclusions

Cancer cells are capable of transitioning between a state of stemness and non-stemness in response to different stimuli from the microenvironment, genetic or epigenetic alterations, and the activation of cellular development programs such as EMT. These alterations confer to cancer stem cells the ability to self-renew, present a high tumorigenic capacity, express key molecules to maintain stem identity, increase drug resistance, and enhance their ability to migrate to form new tumors. Recently, it has been shown that lncRNAs are able to participate in the regulation of the stem phenotype. Transcripts such as LINC00511, HOTAIR, H19, and Lnc304a, among others, have been shown to be key players in the modulation of these capacities. These studies indicate that lncRNAs could represent an interesting target for therapy. The understanding of the mechanisms involved in the regulation of stemness and its repercussions at the biological and clinical level in cancer could lead to the development of new treatments and, probably, with this, to the reduction of mortality caused by malignant tumors.

## Figures and Tables

**Figure 1 cells-11-02352-f001:**
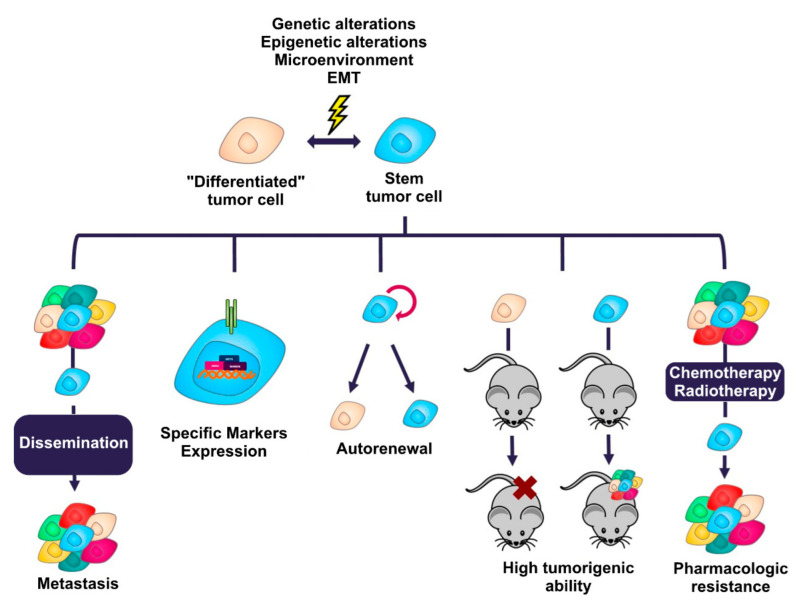
Characteristics that define the stem phenotype in cancer. Cancer cells can transition between a state of stemness and non-stemness generated by multiple genetic and epigenetic alterations, the tumor microenvironment, and the activation of signaling programs such as EMT. These processes allow cancer cells to acquire the ability to metastasize, express membrane proteins or specific intracellular molecules, self-renew, have a high tumorigenic capacity in vivo, and generate resistance to many conventional cancer treatments.

**Figure 2 cells-11-02352-f002:**
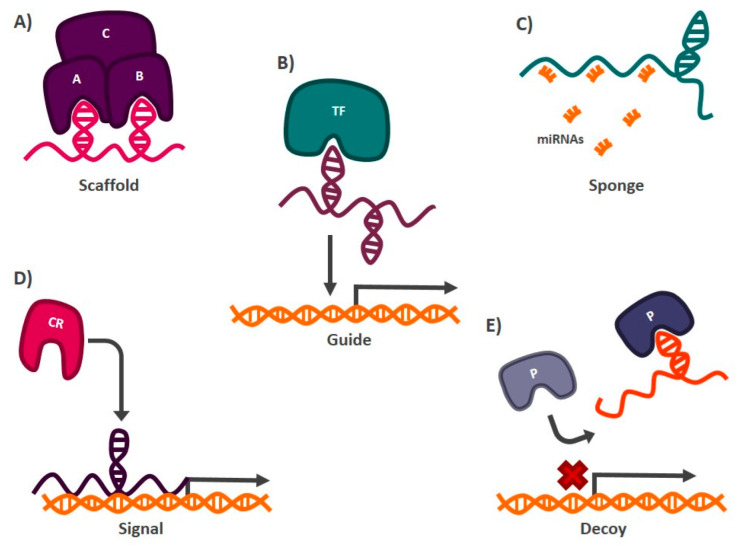
Mechanisms of action of lncRNAs. (**A**) Scaffolds. They allow the formation of protein complexes. (**B**) Guides. They direct gene expression regulatory proteins or transcription factors (TFs) to sites in the genome where activation or inactivation of gene expression is demanded. (**C**) Sponges. They inhibit the activity of miRNAs by interacting directly with them. (**D**) Signals. They indicate to chromatin remodeling (CR) proteins the region of the genome where gene activation or silencing is required. (**E**). Decoy. They prevent the interaction of proteins (P) with their target by interacting directly with them.

**Figure 3 cells-11-02352-f003:**
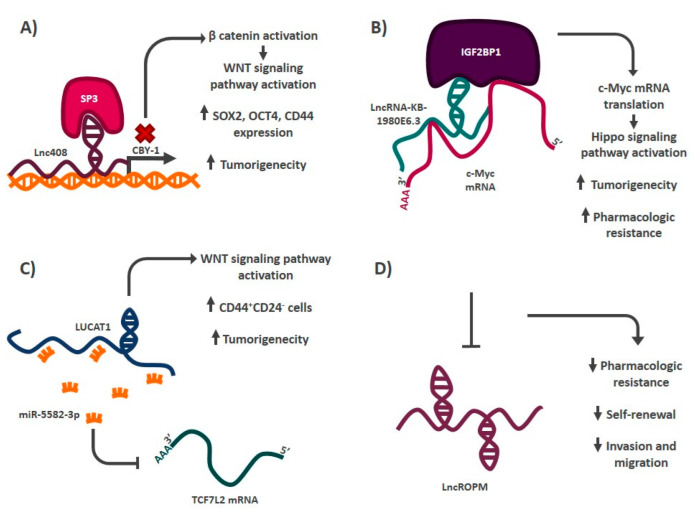
Mechanism of action of some stemness regulatory lncRNAs in breast cancer. (**A**) Lnc408 acts as a guide by recruiting chromatin remodelers to the promoter region of CBY-1, preventing its expression, and allowing the activation of the WNT signaling pathway. (**B**) LncRNA-KB-1980E6.3 forms an RNA–protein complex that stabilizes the mRNA of the C-MYC oncogene, allowing its translation, which generates the acquisition of some characteristics that define the stem phenotype. (**C**) LUCAT1 is a lncRNA that acts as a sponge of miR-5582-3P, promoting the activation of the WNT signaling pathway and thus increasing the proportion of CD44^+^ CD24^−^ cells. (**D**) Inhibition of LncROPM decreases some of the characteristics that define the stem phenotype, although its mechanism has not yet been elucidated.

**Figure 4 cells-11-02352-f004:**
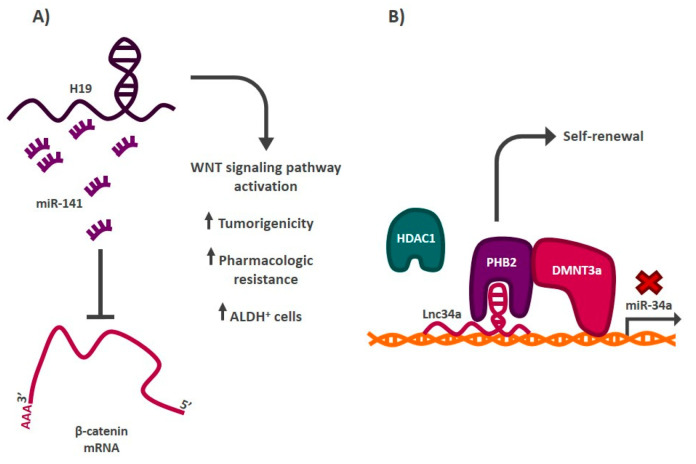
Role of lncRNAs in the regulation of the stem phenotype in colorectal cancer. (**A**) lncRNA H19 functions as a sponge by sequestering miR-141, a negative regulator of β-catenin mRNA, generating the activation of the WNT signaling pathway and increasing the proportion of ALDH cancer stem cells^+^, among other features. (**B**) Lnc34a recruits chromatin remodelers to the promoter region of miRNA-34a, blocking its expression. This allows cancer stem cells to self-renew.

**Figure 5 cells-11-02352-f005:**
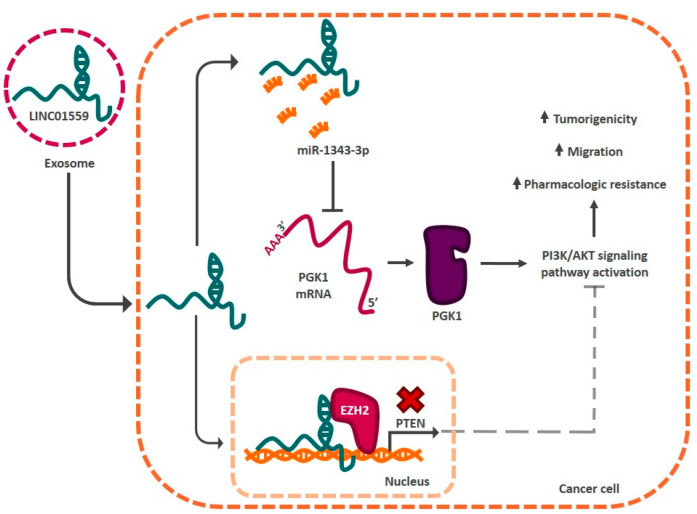
The tumor microenvironment influences the activation of stemness regulatory lncRNAs in gastric cancer. LncRNAs, such as LINC01559, can be secreted by cells in the microenvironment via exosomes. LINC01559 enters cancer cells and activates the PI3K/AKT signaling pathway. There, it acts as a sponge for miR-1343-3p, allowing the translation of PGK1 kinase mRNA and activating the signaling pathway. In addition, this lncRNA acts as a signal to recruit chromatin remodeling complexes to the PTEN promoter region, preventing its expression. This allows the signaling system to be activated and the cancer cells to acquire capacities associated with the stem phenotype.

## Data Availability

Not applicable.

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
