# Peer review of "Regulation of the Cancer Stem Phenotype by Long Non-Coding RNAs"

_cells, 2022, doi:10.3390/cells11152352_

Round 1

Reviewer 1 Report

The work done by Cruz entitled :” Regulation of the cancer stem phenotype by long noncoding 2

RNAs” is well prepared, however it still needs some revision before being accepted for publication,

Please specify more about the mechanism in the section of the introduction.

the authors need to clarify their research question and research hypothesis.

Some English errors, please go over the manuscript 

No information was listed about the previous studies.

Author Response

Thank you for your comment. We modified the article as requested.

The work done by Cruz entitled :” Regulation of the cancer stem phenotype by long noncoding 2

RNAs” is well prepared, however it still needs some revision before being accepted for publication,

Please specify more about the mechanism in the section of the introduction.
We extended the introduction, adding the mechanisms (line 443)

the authors need to clarify their research question and research hypothesis.
The research question was improved (line 45)

Some English errors, please go over the manuscript 
We revised the article

No information was listed about the previous studies.
We are not sure about the section that needs more information, but reviewed the article, adding more references.

Reviewer 2 Report

Overall a good review. The authors have done a good job in gathering the latest knowledge concerning the lncRNA involvement in cancer stemness. I recommend the manuscript for publication in the journal.

I have a suggestion regarding Table 2. In my opinion, the table would be more useful if it is short and only contains the necessary links to the gene, protein or signaling circuit. In the current format there are too much description, most of which are also included in the text. I recommend removing the description part and replacing it with a column mentioning the gene, protein or signaling circuit involvement.

Author Response

I have a suggestion regarding Table 2. In my opinion, the table would be more useful if it is short and only contains the necessary links to the gene, protein or signaling circuit. In the current format there are too much description, most of which are also included in the text. I recommend removing the description part and replacing it with a column mentioning the gene, protein or signaling circuit involvement.

We modified the table accordingly

Reviewer 3 Report

In this review manuscript Jose Adan et al, reviewing regulation of the cancer stem phenotype by long noncoding  RNAs. Long non-coding RNA are shown to be involved in variety of biological function, including cancer progression. In this review, the authors nicely summarized the role of non-coding RNA in stem cell survival, differentiation, drug resistance in several types of cancer. The manuscript is well written and should be published with minor modifications.

Minor modification.

The Cancer stem cells was originally discovered in leukemia by John Dick’s group (ref 4) and search shown many non-coding RNAs are involved in leukemia stem cells phenotype. It would be then appropriate to discuss these papers in this very interesting review.

Author Response

Thank you for your comment. We modified the article as requested.

The Cancer stem cells was originally discovered in leukemia by John Dick’s group (ref 4) and search shown many non-coding RNAs are involved in leukemia stem cells phenotype. It would be then appropriate to discuss these papers in this very interesting review.

We added more information regarding the role of lncRNAs in leukemia stem cells (line 382), with a particular emphasis toward hematopoeitic factors, to keep balanced the review.